# Shared decision making applied to self-management program for hypertensive patients: A scoping review protocol

**So Young Yun[1], Mi Ok Song[2]* **

1 Department of Nursing, Nambu University, Gwangju, South Korea, 2 Department of Nursing, Mokpo National University, Muan-gun, Jeollanamdo, South Korea

☯ These authors contributed equally to this work.
* coffeesong79@gmail.com

**Data Availability Statement:** No datasets were generated or analysed during the current study. All relevant data from this study will be made available upon study completion. All data generated or

## Abstract

Shared decision making (SDM) improves health outcomes by providing individualized nursing interventions to patients and educating and empowering them to actively participate in health. However, hypertensive patients who require self-management do not actively participate in the SDM process. This scoping review aims to investigate the available information on SDM to self- management programs for hypertensive patients. The proposed scoping review protocol will be conducted using the Arksey and O'Malley methodological framework, following the Joanna Briggs Institute's methodology for scoping reviews. Multiple databases will be searched, including the MEDLINE (PubMed), Embase, Cochrane Library. Papers' titles, abstracts, and full text will predominantly be screened by one researcher with a second researcher. Data will be extracted using a customized table developed in collaboration with the research team, and presented in tabular format, arranged thematically. In the scoping review, studies involving the self- management program applied for hypertensive patients with two or more of the components of the intervention are included. And in the SDM process, studies in which healthcare providers provide various options for patients' self- management and set health goals in consideration of patients' preferences and values will be included.

## Introduction

Shared decision-making (SDM) refers to the process by which at least two participants share information and collaborate to reach an agreement on the preferred treatment of both parties [1]. SDM is a process in which decisions are made in a collaborative manner, where reliable information is provided along with alternative options, and consensus is reached that considers the concerns, values, goals, and preferences of patients and their family [2]. Recently, the patient's participation in decision-making has been emphasized in the process of determining treatment methods, and the informed preferences of patients and their family should play the most important role in the decision-making process [3]. Many organizations around the world are recommending the use of SDM in decision-making in healthcare situations [4, 5].

analyzed in this study will be included in the published scoping review article.

**Funding:** This study was supported by research funds from Nambu University, 2022 The funders had no role in study design, data collection and analysis, decision to publish, or preparation of the manuscript.

**Competing interests:** The authors have declared that no competing interests exist.

Self- management is a process of maintaining health through health promotion activities and disease management [6], and hypertensive patients need to adapt to self- management behaviors. Self-management for hypertensive patients includes medication, lifestyle management, and regular hospital visits, etc. [7]. Nurses provide self- management support and decision-making support to patients through these self- management interventions and allow patients to participate in the self- management process and obtain information. SDM is also applied to this process, and by encouraging patient participation in the self- management process and providing necessary support to patients, it improves the quality of the decision-making process and treatment implementation, ultimately improving patient outcomes [2]. In the recent middle-range theory of nursing in hypertensive patients, the concept of SDM was presented to the nursing management process for lifestyle changes in hypertensive patients, and problem-solving through mutual goal setting was emphasized [7]. As such, it is necessary to apply SDM to self- management programs for hypertensive patients (SMPHP).

SDM strengthens patients' autonomy by providing individualized nursing interventions to patients and taking responsibility for their daily decisions about health problems [8]. In addition, educating and empowering patients to actively participate in their health, improves health outcomes such as compliance with medication [9, 10], overall health status, disease knowledge, and improvement in lifestyle [10]. However, there are concerns that hypertensive patients may not actively participate in the SDM process [11, 12]. Both healthcare providers and patients experience stress about participating in the SDM process for reasons such as unfamiliarity with SDM [12], problems with old age and health literacy, and differences in knowledge between healthcare providers and patients [13].

In this respect, it is necessary to discuss how to effectively apply SDM to SMPHP. To increase the participation of hypertensive patients in decision-making, it is necessary to consider how to form a supportive relationship with medical service providers, motivate patients, and provide accurate and sufficient health information [14]. Regarding the method specifically applied in the SDM process, it would be systematic to consider the three-talk model of SDM as a standard. In the model, the SDM process is explained by dividing it into 'team talk', 'option talk', and 'decision talk' [2].

Hypertension causes various cardiovascular diseases, increases premature mortality, and imposes a significant burden on individuals, family, and society. In Korea, the prevalence of hypertension in those aged 19 and older increased from 23.7% in 2010 to 29.0% in 2020, but the control rate was 47.6%, which has not changed significantly over the past 10 years [15]. In this context, the government is implementing a hypertension registration management project, and each public health center is implementing self-management intervention through health education, individual phone counseling, and visiting health services. However, the application of SDM to SMPHP is insufficient [14]. An integrative review that identified the comprehensive process of SDM in healthcare settings [16] and a systematic literature review of the effects of SDM applied to hypertensive patients were conducted [17], but there was no study on the topic of applying SDM to SMPHP.

There are currently no scoping review papers summarizing existing research on this topic to help identify evidence and methodological differences. Therefore, the scoping review to contrast and discuss existing studies in this field is meaningful due to differences between previous studies, various methodologies, patient populations, inclusion and exclusion criteria. Due to the heterogeneity and scope of available literature, scoping review are acceptable methodologies that can be used rather than systematic reviews that require a narrower selection of research questions and quality evaluation papers [18].

## Aims

The aims of the proposed ScR are:

1. To summarize the current evidence on the SDM as it relates to SMPHP.

2. To identify existing gaps to inform future research in this area.

## Materials and methods

The proposed scoping review will be conducted using the Arksey and O'Malley methodological framework [18] and follows the Joana Briggs Instituted (JBI) methodology for scoping reviews [19]. A checklist for the development and reporting of this ScR protocol [20] is included within the supplementary material (S1 Checklist) and a full scoping review will be reported according to PRISMA extension for Scoping Review (PRISMA-ScR) [21].

### Stage 1: Identifying the research questions

**Research question.** The scoping review question is as follows.

1. How have SDM progresses been applied to SMPHP in various healthcare environments?

2. What methods were used to actively involve the subject in the SDM process?

3. Are there any considerations in the SDM process for the effectiveness of the self- management program?

Types of participants. Studies involving patients aged 19 years or older with a diagnosis of hypertension will be selected. Hypertensive patients with other comorbidities will be selected, but studies involving secondary hypertensive patients will be excluded.

*Concept.* The SMPHP in the study will be selected only if two or more of the components of the intervention (medication, diet, physical activity, BP monitor, smoking cessation, etc.) are included. And in the SDM process, studies in which healthcare providers provide various options for patients' self-management and set health goals in consideration of patients' preferences and values will be included.

*Context.* In the scoping review, SMPHP that have been implemented in any healthcare environment, whether in an acute hospital setting or a community primary healthcare setting, will be selected, and the occupational group of interventionists will not be limited.

*Type of sources.* In the literature selection process, special research designs will not be restricted, but qualitative studies, protocols, opinion papers, and letters that cannot confirm specific components of a self-management program and the progress of SDM will be excluded.

### Stage 2: Identifying relevant studies

Multiple databases were searched in January 2023, including the MEDLINE (PubMed), Embase, Cochrane Library. Following JBI review methods, a three-step search strategy was employed. Initially, a limited search was conducted on the research topic in MEDLINE (PubMed) and Embase database. Subsequently, the text words from titles and abstracts of retrieved articles, along with index terms used to describe the articles, were analyzed. A second search was conducted on MEDLINE (PubMed), Embase, and Cochrane Library using all identified keywords and index terms. Additionally, a search for gray literature was conducted on Google Scholar. Thirdly, additional sources will be searched by examining the reference list of the finally selected literature. An example search strategy for MEDLINE (PubMed) is

presented in the supplementary materials of this protocol (S1 File). There are no restrictions on language or year of publication.

## Stage 3: Study selection

After the literature search, the identified documents are uploaded using EndNote X8 8.2 (Clarivate Analytics, Philadelphia, PA, USA) and duplicates are removed. Titles and abstracts are independently will be reviewed by two reviewers to evaluate inclusion and exclusion criteria, and the literature would be first selected. Two reviewers independently will evaluate the entire contents of the initially selected literature to select the literature for final analysis. Any disagreements that arise during the selection process should be discussed to reach a consensus. The literature search and selection process will be presented in the PRISMA-ScR Flowchart [21].

Inclusion criteria will comprise adults (aged 19 years and above) diagnosed with Hypertension (HTN), and a self-management intervention encompassing a minimum of two or more intervention components (e.g., education, physical activity, pharmacotherapy, or lifestyle modification). Exclusion criteria will entail secondary and tertiary data studies, abstracts, or protocol-only papers, and inaccessible full texts. SDM intervention is expected to entail an iterative process whereby healthcare providers and participants collaboratively establish goals. Studies where the participant assumes the role of the healthcare provider will be excluded. Before the team begins source selection, a pilot test is conducted with a random sample of 25 titles/abstracts. Screening begins when 75% consent of the team is achieved.

## Stage 4: Charting the data

**Data extraction.** Data will be extracted using a data extraction tool developed by the researcher based on the data extraction part of the JBI scoping review [18]. The components of the self-management program and the process of SDM will be extracted along with the general characteristics of the literature. The SDM process is divided into team talk, option talk, and decision talk [2] to extract data. The details of the extraction tool are shown in Table 1. Before data extraction, a pilot test will be conducted using the extraction tool for the two selected literature, and then it was confirmed that all relevant results were extracted. The data extraction tool can be modified during the data extraction process and will be written in such a way that one reviewer extracts independently and the other reviewer verifies it. Any disagreement between reviewers will be resolved through discussion during data extraction process.

**Table 1. Data extraction tool.**

| Categories | Sub-categories | Details |
|---|---|---|
| General characteristics | Authors<br>Year of publication<br>Country of origin<br>Study design | |
| Self-management program | Setting<br>Participants<br>Contents<br>Theory | |
| How shared decision making was applied | Three-talk model | Team talk |
| | | Option talk |
| | | Decision talk |
| | Decision aid | |
| | | |
| Outcomes | | |

### Stage 5: Collating, summarizing, and reporting the results

**Data analysis and presentation.** Data analysis will be presented by analyzing the results produced by the extraction tool. A descriptive analysis of the general characteristics of the literature and an analysis of research questions will be conducted. Analysis based on the theory of the SDM process will be used when applying the SMPHP in the future, and the three-talk model and implementation of the SDM process will be compared. Quantitative data will use descriptive statistics, and descriptive summaries and tabular results will be used to characterize SMPHP. The results of this study will reveal strategies for applying SDM when planning a SMPHP.

### Ethical consideration

This study will begin analysis after receiving approval of the application for exemption from deliberation by the Institutional Review Board of Nambu University (IRB #1041478-2021-HR-035).

### Strengths and limitations of this study

- The review will be one of the first study to examine the SDM process of SMPHP.

- Exploring this issue has the potential to highlight the areas that impact on the intervention of SMPHP.

- It would also inform healthcare providers about the challenges and the needs for SDM of SMPHP so that support and resources are directed to increase effectiveness.

### Registration

This project has been created for the Scoping Review on the OSF repository and the protocol and search strategies uploaded. The project has also been registered with the OSF Registries.

- OSF project: https://osf.io/rtfvq

- DOI: 10.17605/OSF.IO/D8SPM

### Supporting information

**S1 Checklist. PRISMA-P 2015 checklist.** This Checklist is used for reporting scoping review protocol.
(DOCX)

**S1 File. Sample search strategy for MEDLINE (PubMed).**
(DOCX)

### Acknowledgments

We would like to thank those who conducted the study for SDM processes implemented within SMPHP so that this study could be conducted.

### Author Contributions

**Conceptualization:** So Young Yun, Mi Ok Song.

**Data curation:** So Young Yun, Mi Ok Song.

**Methodology:** So Young Yun, Mi Ok Song.

**Supervision:** So Young Yun.

**Writing – original draft:** So Young Yun, Mi Ok Song.

**Writing – review & editing:** Mi Ok Song.

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
