## [Decision Letter · Decision Letter 0]

3 Apr 2024

PONE-D-23-31258Shared decision making applied to self-management program for hypertensive patients: A scoping review protocolPLOS ONE

Dear Dr. Song,

Thank you for submitting your manuscript to PLOS ONE. After careful consideration, we feel that it has merit but does not fully meet PLOS ONE’s publication criteria as it currently stands. Therefore, we invite you to submit a revised version of the manuscript that addresses the points raised during the review process.

We look forward to receiving your revised manuscript.

Kind regards,

Saliha Karadayi-Usta, PhD

Academic Editor

PLOS ONE

Journal Requirements:

"This study was supported by research funds from Nambu University, 2022"

Reviewers' comments:

Reviewer's Responses to Questions

**Comments to the Author**

1. Does the manuscript provide a valid rationale for the proposed study, with clearly identified and justified research questions?

Reviewer #1: Yes

Reviewer #2: Yes

2. Is the protocol technically sound and planned in a manner that will lead to a meaningful outcome and allow testing the stated hypotheses?

Reviewer #1: Yes

Reviewer #2: Yes

3. Is the methodology feasible and described in sufficient detail to allow the work to be replicable?

Reviewer #1: Yes

Reviewer #2: No

4. Have the authors described where all data underlying the findings will be made available when the study is complete?

Reviewer #1: Yes

Reviewer #2: Yes

5. Is the manuscript presented in an intelligible fashion and written in standard English?

Reviewer #1: Yes

Reviewer #2: Yes

6. Review Comments to the Author

You may also provide optional suggestions and comments to authors that they might find helpful in planning their study.

Reviewer #1: Yun SY and Song MO investigate the available information on shared decision-making (SDM) in self-management programs for hypertensive patients. This is because SDM improves health outcomes by providing personalized nursing interventions to patients and educating and empowering them to become active participants in their health, but hypertensive patients who need to self-manage do not actively participate in the SDM process.

They are planning to use multiple databases, including MEDLINE (PubMed), Embase, Cochrane Library, and Google Scholar. In the scoping review, studies involving the self-management program applied for hypertensive patients with two or more of the components of the intervention are included.

I believe that this study is well-planned and is expected to yield very important results.

Reviewer #2: Dear Authors

After reviewing your manuscript I have to ask the following questions:

You indicate that google Scholar is a database where you are going to carry out your study, I must point out that google scholar is not a scientific database, but a generalist search engine.

Furthermore, it would be appropriate to describe in a clearer way the process of search and selection of documents to be included in the study. After reading stage 2, it is not easy to really understand the meaning of the first search and even the use of Gogle Schola for gray literature search.

Finally, regarding the document selection criteria, it is necessary to specify in greater detail the elements that will help in the selection of documents.

7. PLOS authors have the option to publish the peer review history of their article (what does this mean?). If published, this will include your full peer review and any attached files.

Reviewer #1: No

Reviewer #2: No

---

## [Author Response · Author response to Decision Letter 0]

18 Jun 2024

Dear Editor and Reviewer:

On behalf of my-co-authors, I thank you and the reviewers for your thoughtful comments and recommendation. We have made the following changes to the manuscript to address your concerns. The revisions are shown in Red in the text. 

Response to Editor`s Comments

#1. Please ensure that your manuscript meets PLOS ONE's style requirements, including those for file naming. 

=> According to the editor's opinion, manuscript formatting, Title page and file naming have been revised. 

#2. Thank you for stating the following financial disclosure:

"This study was supported by research funds from Nambu University, 2022"

=> According to the editor's opinion, Authors stated what role the funders took in the study. It is state in the cover letter that "The funders had no role in study design, data collection and analysis, decision to publish, or preparation of the manuscript." 

#3. Please provide a complete Data Availability Statement in the submission form, ensuring you include all necessary access information or a reason for why you are unable to make your data freely accessible. If your research concerns only data provided within your submission, please write "All data are in the manuscript and/or supporting information files" as your Data Availability Statement.

=> According to the editor's opinion, it was stated that " No datasets were generated or analyzed during the current study. All relevant data from this study will be made available upon study completion." for Data Availability Statement in the submission form. 

#4. When completing the data availability statement of the submission form, you indicated that you will make your data available on acceptance. We strongly recommend all authors decide on a data sharing plan before acceptance, as the process can be lengthy and hold up publication timelines. Please note that, though access restrictions are acceptable now, your entire data will need to be made freely accessible if your manuscript is accepted for publication. This policy applies to all data except where public deposition would breach compliance with the protocol approved by your research ethics board. If you are unable to adhere to our open data policy, please kindly revise your statement to explain your reasoning and we will seek the editor's input on an exemption. Please be assured that, once you have provided your new statement, the assessment of your exemption will not hold up the peer review process.

=> According to the editor's opinion, all authors decided on a data sharing plan before acceptance and completed the data availability statement of the submission form 

Response to Reviewer #2`s Comments

#1. You indicate that google Scholar is a database where you are going to carry out your study, I must point out that google scholar is not a scientific database, but a generalist search engine. Furthermore, it would be appropriate to describe in a clearer way the process of search and selection of documents to be included in the study. After reading stage 2, it is not easy to really understand the meaning of the first search and even the use of Google Schola for gray literature search.

=> According to the reviewer's opinion, the literature search process had been revised and suggested in line 114~122 of the manuscript. 

Multiple databases were searched in January 2023, including the MEDLINE (PubMed), Embase, Cochrane Library. Following JBI review methods, a three-step search strategy was employed. Initially, a limited search was conducted on the research topic in MEDLINE (PubMed) and Embase database. Subsequently, the text words from titles and abstracts of retrieved articles, along with index terms used to describe the articles, were analyzed. A second search was conducted on MEDLINE (PubMed), Embase, and Cochrane Library using all identified keywords and index terms. Additionally, a search for gray literature was conducted on Google Scholar. Thirdly, additional sources will be searched by examining the reference list of the finally selected literature.

#2 Finally, regarding the document selection criteria, it is necessary to specify in greater detail the elements that will help in the selection of documents.

=> According to the editor's opinion, the document selection criteria were revised at lines 135-141 and exclusion criteria were presented in detail. 

Inclusion criteria will comprise adults (aged 19 years and above) diagnosed with Hypertension (HTN), and a self-management intervention encompassing a minimum of two or more intervention components (e.g., education, physical activity, pharmacotherapy, or lifestyle modification). Exclusion criteria will entail secondary and tertiary data studies, abstracts, or protocol-only papers, and inaccessible full texts. SDM intervention is expected to entail an iterative process whereby healthcare providers and participants collaboratively establish goals. Studies where the participant assumes the role of the healthcare provider will be excluded.

---

## [Decision Letter · Decision Letter 1]

15 Aug 2024

Shared decision making applied to self-management program for hypertensive patients: A scoping review protocol

PONE-D-23-31258R1

Dear Dr. Song,

We’re pleased to inform you that your manuscript has been judged scientifically suitable for publication and will be formally accepted for publication once it meets all outstanding technical requirements.

Kind regards,

César Leal-Costa, Ph. D

Academic Editor

PLOS ONE

Additional Editor Comments (optional):

Reviewers' comments:

Reviewer's Responses to Questions

**Comments to the Author**

1. Does the manuscript provide a valid rationale for the proposed study, with clearly identified and justified research questions?

Reviewer #1: Yes

2. Is the protocol technically sound and planned in a manner that will lead to a meaningful outcome and allow testing the stated hypotheses?

Reviewer #1: Yes

3. Is the methodology feasible and described in sufficient detail to allow the work to be replicable?

Reviewer #1: Yes

4. Have the authors described where all data underlying the findings will be made available when the study is complete?

Reviewer #1: Yes

5. Is the manuscript presented in an intelligible fashion and written in standard English?

Reviewer #1: Yes

6. Review Comments to the Author

You may also provide optional suggestions and comments to authors that they might find helpful in planning their study.

Reviewer #1: The authors appear to have responded appropriately to the reviewer's comments. Therefore, I recommend this paper for acceptance.

7. PLOS authors have the option to publish the peer review history of their article (what does this mean?). If published, this will include your full peer review and any attached files.

Reviewer #1: No

---

## [Editor Report · Acceptance letter]

24 Sep 2024

PONE-D-23-31258R1 

PLOS ONE

Dear Dr. Song, 

I'm pleased to inform you that your manuscript has been deemed suitable for publication in PLOS ONE. Congratulations! Your manuscript is now being handed over to our production team.

Kind regards, 

on behalf of

Dr. César Leal-Costa 

Academic Editor

PLOS ONE